

# Benchmarking a fast, satisficing vehicle routing algorithm for public health emergency planning and response: "Good Enough for Jazz"

Emma L. McDaniel[1,2], Sampson Akwafuo[3], Joshua Urbanovsky[4] and Armin R. Mikler[1,2]

[1] Department of Computer Science, Georgia State University, Atlanta, Georgia, United States of America
[2] Center for Disaster Informatics and Computational Epidemiology, Georgia State University, Atlanta, Georgia, United States of America
[3] Department of Computer Science, California State University, Fullerton, California, United States of America
[4] Unaffiliated, Seattle, Washington, United States of America

## ABSTRACT

Due to situational fluidity and intrinsic uncertainty of emergency response, there needs to be a fast vehicle routing algorithm that meets the constraints of the situation, thus the receiving-staging-storing-distributing (RSSD) algorithm was developed. Benchmarking the quality of this satisficing algorithm is important to understand the consequences of not engaging with the NP-Hard task of vehicle routing problem. This benchmarking will inform whether the RSSD algorithm is producing acceptable and consistent solutions to be used in decision support systems for emergency response planning. We devise metrics in the domain space of emergency planning, response, and medical countermeasure dispensing in order to assess the quality of RSSD solutions. We conduct experiments and perform statistical analyses to assess the quality of the RSSD algorithm's solutions compared to the best known solutions for selected capacitated vehicle routing problem (CVRP) benchmark instances. The results of these experiments indicate that even though the RSSD algorithm does not engage with finding the optimal route solutions, it behaves in a consistent manner to the best known solutions across a range of instances and attributes.

## INTRODUCTION

The problem addressed in this article is motivated by the need to calculate routes for the delivery of medical countermeasures (MCMs) to points of dispensing (PODs) during a bio-emergency. This problem is characterized by situational fluidity and intrinsic uncertainty of disaster response. We have implemented and evaluated an algorithm for solving a specific instance of a capacitated, multi-vehicular routing problem. The algorithm described in this article is a revised version of one that was originally proposed in a dissertation by *Urbanovsky (2018)*. Henceforth, we will call the algorithm discussed in this article RSSD (Receiving-staging-storing-distributing algorithm). The rational for

Corresponding author
Emma L. McDaniel,
emcdaniel10@gsu.edu

using RSSD lies in the need to re-compute delivery routes quickly in frequent succession to explore solutions under changing conditions. RSSD utilizes a satisficing problem criteria and does not attempt to optimize. For problems that require solutions that are in compliance with stated constraints, the RSSD algorithm generates solutions quickly as it does not attempt to optimize the solution once a compliant solution is found. In this work, we describe RSSD and quantify the consequences of RSSD not engaging with the NP-Hard nature of the vehicle routing problem by benchmarking RSSD against best known solutions.

Regional public health preparedness planners (PHPPs) are tasked with creating plans for the delivery of MCMs to their regions' population in the case of a bio-emergency, for instance the accidental or deliberate release of anthrax. Current plans use the POD model, where there are pre-selected locations within the public health region, at which MCMs are dispensed to the public (*Abbey, Aaby & Herrmann, 2013*; *Gorman, 2016*). Each POD serves a corresponding sub-population represented by a set of population blocks in proximity to its location. The demand for MCMs are the sum of the population of the block groups identified as served by the POD, the population count data are obtained from the most recent United States' Decennial Census or American Community Survey (*Jimenez, Mikler & Tiwari, 2012*; *O'Neill et al., 2014*). Once an emergency has been declared, the United States federal government makes available MCM resources, which are delivered to a predetermined regional receiving, staging, and storing (RSS) site/s (*i.e.*, depot/s).

Given a specific time to complete delivery, the feasibility of the solution is a function of the number and capacity of available vehicles. Hence, this problem represents an instance of a capacitated, multi-vehicular routing problem. Vehicle routing or travelling salesman problems are known to be NP-Hard, and coupled with situational fluidity and uncertainty of the problem, the complexity and its upper bound become more confounded. In disaster response, where time and performance capabilities are limited, it is often preferable to work towards a feasible or satisficing set of MCM distribution routes, rather than to engage with an NP-Hard problem of finding the optimal. *Satisficing* is a term famously coined by *Simon (1956)*; he purports that it is rational to change the bounds of a complex cost function from searching for the optimal to one that *satisfices* the constraints appropriately.

When given the capacity of vehicles and the available delivery time, RSSD computes satisficing, feasible routes. Each of those routes satisfice the given constraints, *i.e.*, "good enough for jazz." The article will focus on the comparison to known best solution sets (*Uchoa et al., 2017*) in order to determine the level of quality of solutions produced by RSSD. In what follows, we will describe RSSD and benchmark it to assess the consequences of a *satisficing* algorithm. The article is organized as follows: first, the 'Background' sections present an overview of the domain space of RSSD and a 'Related Works' section. Following this is the 'Problem Description', which describes the algorithm and benchmarking methods utilized. Afterwards, we present the 'Results' of the benchmarking RSSD experiments; and lastly, the 'Conclusion' contains the implications of the results, limitations, and possible future works.

## BACKGROUND

To describe RSSD with an American idiom, it is "good enough for jazz," or good enough to suit the purposes of the situation, while knowing that the results will be suboptimal. To fill a specific niche within emergency planning, response, and medical countermeasure delivery, a version of this capacitated, multi-vehicular routing algorithm described in this article was formulated in previous work by *Urbanovsky (2018)*. This RSSD algorithm was utilized in a future iteration of the REsponse PLan ANalyzer (RE-PLAN), a decision support system for emergency planning and response, as described in *O'Neill et al. (2014)*. All of RE-PLAN's versions and modules were developed in collaboration with the Texas department of public health and have been used for planning activities in public health regions in Texas and California (*O'Neill, Poole & Mikler, 2021*).

In 2004, through the cities readiness initiative (CRI), many regional public health departments across the United States were mandated to create plans for dispersing preventative antibiotics to the public in the case of a large-scale accidental or intentional release of biochemicals (*Nelson et al., 2010*; *Avchen, LeBlanc & Kosmos, 2018*). One of the modes of dispensing proposed by CRI is the POD model. The *planning stage* for the POD model is pre-incident and is often revisited with an audit. The *activation stage* is in the event of an accidental or deliberate release of biochemicals. For both stages, the RSSD algorithm must meet certain conditions that are unique to each stage.

In the planning stage, PHPPs must calculate the demands for the PODS, create preliminary routes from depot/s to PODs, and identify the number of vehicles needed. With POD demands, identified POD and RSS locations, a time constraint, and a capacity constraint, the RSSD algorithm generates routes and identifies the number of vehicles necessary. The identification of the number of vehicles needed is essential in this planning stage. See Fig. 1 for a timeline of the planning stage. It is clear in this timeline that there is revisiting of steps in order to improve the plan. Thus, it is conducive to planning that the routing algorithm allows for quick and consecutive creation of solutions.

In the activation stage, the timeline of events for a response is stochastic see Fig. 1. There is a 96-h window from a release of biochemicals to when MCMs must be completely dispensed to the public. At "hour 0", an incident occurs; and depending on the type of incident, the discovery can be instantaneous or, if the incident is insidious in nature, can take as long as the onset of symptoms and subsequent diagnosis/es from medical professionals. The time to detect the release is unknown and is indicated on the timeline with a "?" and a dotted line. It is essential specifically for anthrax, for those impacted to receive the antibiotics soon after exposure, otherwise, the antibiotics will have minimal impact on the course of infection. Without the distribution of MCMs quickly, hospitalization for those exposed will be necessary to neutralize the bacterial infection and could result in overcrowding of emergency rooms.

Once the event is characterized as a release, the public health department will inform the strategic national stockpile (SNS) that such an event has occurred and the region needs to be pushed MCMs. The SNS will then work to deliver the MCMs needed to the region's central depot(s) within 12 h of notification. The length of time it takes for the SNS to

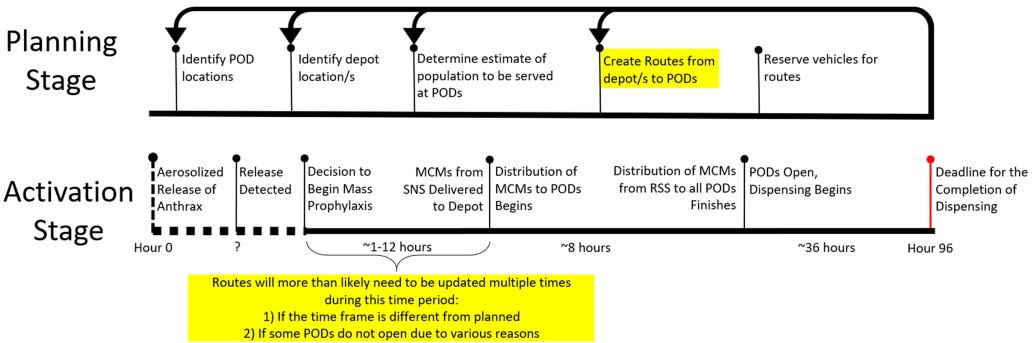

**Figure 1 Timeline of response and activation stages in the case of an accidental or deliberate release of anthrax.** The highlighted portions of the timeline represent the expected times at which the RSSD algorithm will be run to generate time and capacity compliant solutions. The algorithm may be run multiple times during both the planning and activation stages.

deliver the MCMs to the regional depot is also dynamic and is dependent on the undisclosed location/s of SNS' warehouses and the risk assessment of such an event in that public health region (*Center for Disease Control, 2014*; *Neumeister & Gray, 2021*). Therefore, in order to utilize the time available effectively, it is potentially necessary to reroute with a decreased or increased time constraint dependent on the time remaining.

Due to unforeseen circumstances, some PODs from the initial plan are unable to be activated leaving the planned routes unusable. This then requires an adjustment to the demand served at PODs, and a recalculation of the routes. The step at which consecutive runs of RSSD might happen is highlighted in the Activation Stage of Fig. 1. This portion of the timeline is unpredictable, therefore, a routing algorithm that produces feasible solutions quickly is essential.

Both the planning stage and activation stage require quick solutions because of the need to rerun routes based on a variety of factors. Most public health departments do not have access to adequate computing infrastructure with the capabilities to run complex optimizers quickly. With all the constraints that the algorithm must perform under, the choice to use a greedy algorithm with a couple of metaheuristics stems from avoiding engagement in this NP-Hard problem; instead, the focus is to ensure the routes created meet the specified time constraint and capacity constraint. The RSSD algorithm's solutions are feasible routes within these constraints and satisfice the needs of PHPPs in both the planning and activation stage.

## Related work

The unpredictable nature of disaster situations requires a modified approach to the capacitated vehicle routing problem (CVRP) to include a number of context specific constraints that are inherently variable and often only loosely defined. This can present some unique challenges within the vehicle routing problems (VRPs) within emergency planning, response, and MCM distribution. During our literature review, we were unable to find any algorithms that matched the same criteria or solved the exact same problem as the RSSD algorithm; however, we did discover algorithms in a similar application domain.

**Table 1 A selection of similar algorithms to RSSD within emergency planning and response.**

| Attributes/Algorithm | Shen, Dessouky & Ordóñez (2009) and Shen, Ordóñez & Dessouky (2009) | Qin et al. (2017) | Gharib et al. (2018) | Li & Chung (2018) | Goli & Malmir (2020) | RSSD algorithm |
|---|---|---|---|---|---|---|
| Unknown number of vehicles | ✗ | ✗ | ✗ | ✗ | ✗ | ✓ |
| Capacity constraint | ✓ | ✗ | ✗ | ✓ | ✓ | ✓ |
| Time constraint | ✓ | ✓ | ✓ | ✓ | ✗ | ✓ |
| Multi-depot | ✗ | ✓ | ✓ | ✓ | ✗ | ✓ |
| Heterogeneous fleet | ✗ | ✗ | ✓ | ✗ | ✓ | ✗ |
| Assume unknown demand | ✓ | ✓ | ✗ | ✗ | ✓ | ✗ |
| Prioritize specific nodes | ✗ | ✗ | ✓ | ✗ | ✓ | ✗ |
| Attempting to optimize | ✓ | ✓ | ✓ | ✓ | ✓ | ✗ |
| Satisficing strategy | ✗ | ✗ | ✗ | ✗ | ✗ | ✓ |

See Table 1 for the strengths and weaknesses of some of these algorithms and RSSD algorithm. Table 1 contains check marks and × marks to indicate some of the unique aspects of the chosen algorithms. Brief descriptions of the algorithms and article contributions are included below.

Shen, Dessouky & Ordóñez (2009) and Shen, Ordóñez & Dessouky (2009), in two articles, propose a mix integer heuristic model with a focus of mitigating some uncertainties in the road network. Their assumptions include: (1) that the demands and time between PODs are unknown at the time of plan creation and, (2) when PODs are activated the demands at each POD location will not be met by the MCMs delivered. As a result of these assumptions, they derived a two-step algorithm approach for the planning stage and the operational phases. In the planning stage, their first algorithm optimally solves the routes (Shen, Dessouky & Ordóñez, 2009) and in the activation stage, they use a stochastic routing algorithm that optimizes the amount of demand that is not met (Shen, Ordóñez & Dessouky, 2009). Qin et al. (2017) propose an update to a Genetic Algorithm utilizing a single-depot VRP after an emergency, where supplies are limited. They attempt to minimize the distance and number of vehicle costs. Gharib et al. (2018) propose an artificial neural fuzzy inference system of clustering the demand points by crisis severity and other criteria. Once these clusters are formed, the points in the cluster are prioritized. They then used two commodity distribution models, MO Firefly and NSGA-II, for their heterogeneous fleet and multi-depot problem. Li & Chung (2018) purport that stochastic programming may not work in emergency situations and therefore deterministic models like CVRP and SDVRP are more robust in applications where the probability distribution is unknown. They propose a robust CVRP and SDVRP using robust optimization techniques to be used in disaster relief routing. Goli & Malmir (2020) use a covering tour approach with fuzzy demand to minimize the arrival time of the last vehicle to nodes. They utilize a mathematical model solved by credibility theory and harmony search algorithm to

decide which points are not mandatory, using this they are able to send vehicles to nodes that have the most need.

The described algorithms create solutions that deliver products from a depot to nodes in emergency situations with time and/or capacity constraints. These algorithms, including RSSD, have formulated the VRP differently in each case. As a result, we determined that it would not be adequate to directly compare algorithms and their solutions due to the differences in the problems being solved. We chose to benchmark the solutions of RSSD against best known solutions for CVRP, treating these best known solutions as a form of ground truth.

### Satisficing and "Good enough" approaches in VRP

Routing literature that uses the satisficing, "good enough," or bounded rationality frameworks are often focused on optimizing how much a routing algorithm can violate constraints and still be feasible or optimize "enough." This type of variant of VRP is described by *Oyola, Arntzen & Woodruff (2018)* as within the problem space of stochastic vehicular routing and chance-constrained routing. *Vidal, Laporte & Matl (2020)* characterize this type of VRP as deterministic with a focus on reliability and goal of mitigating risk of failure. *Jaillet, Qi & Sim (2016)* propose an alternate method of utilizing uncertainty while routing with a violation index that allows routes to satisfice or partially meet constraints by a certain threshold, and thus satisfice the needs of the delivery company. *Nguyen et al. (2016)* build upon *Jaillet, Qi & Sim (2016)* by using the satisficing index in relation to potential customer dissatisfaction with the lateness of delivery. The use of "satisficing" or focus on "good enough" in these VRPs formulations are how much not to optimize. In contrast to these VRP formulations that focus on how much not to optimize, our approach involves changing the overall objective of the algorithm to produce solutions that are sufficient, satisfactory, or *satisficing*.

### Benchmark dataset repositories

Many VRP benchmark datasets exist, often tailored to specific applications or variants of VRP. Research proposing new VRP algorithms frequently use custom datasets to demonstrate their algorithms' performance. While it is important to understand the algorithm in its application space, it is also valuable to compare to other solutions so to evaluate quality.

A popular VRP dataset repository is VRP-REP, which is an open-data platform and reported in *Mendoza et al. (2014)*. When this article was being developed, there were 87 references to this repository, and 32,124 downloads of datasets. *Gunawan et al. (2021)* give an extensive literature review of the VRP benchmark datasets that are available on the VRP-REP website.

Another repository of datasets is the capacitated vehicle routing problem library (CVRPLIB), which was published by *Uchoa et al. (2017)*. This repository contains well-known and often used CVRP datasets since 1959. One of the goals for this repositry as stated by *Uchoa et al. (2017)* was to create a repository of CVRP that all researchers within the field could use when benchmarking their CVRP algorithms. For most instances in this

**Table 2 Symbols and their definitions.**

| Symbol | Definition |
|--------|------------|
| $G$ | Graph, $G$, composed of $G(V, E)$ |
| $V$ | All node and depot/s locations, each with a corresponding demand |
| $n$ | Number of nodes |
| $E$ | Edges between all nodes |
| $D$ | Distance matrix between all $V$ |
| $p$ | $p \subset V$, set of depot/s and do not have a demand |
| $t(r_i)$ | $t$([list of vertices]) returns the sum of the distance between the nodes in explicit order |
| $c(r_i)$ | $c$([list of vertices]) returns the sum of the demand of all nodes in the route |
| $\tau$ | Time constraint |
| $\psi$ | Capacity constraint |

repository, there are a reported known best known route solutions. The provided best known solution has been found by differing CVRP algorithms and this best known solution can help evaluate the results of other CVRP algorithm solutions. We chose to use datasets from CVRPLIB to benchmark our algorithm due to the availability of the best known solutions that we can treat as *ground truth*. We opted to not include datasets from the VRP-REP repository in our assessments for a few reasons: not all have best-known solutions, most datasets have differing formats; and there are specific attributes within the CVRPLIB datasets, which will be discussed later, that we found particularly valuable for our analysis.

# PROBLEM DESCRIPTION

This 'Problem description' section is split into two distinct parts: 'Algorithm overview' and 'Benchmarking methods'. The 'Algorithm overview' subsection contains a brief overview of RSSD, a version of the algorithm that was first published in *Urbanovsky (2018)* and was used within a later version of RE-PLAN that was reported in *O'Neill et al. (2014)*. The 'Benchmarking methods' subsection outlines the methods used in order to evaluate RSSD using one of the datasets in the CVRPLIB repository (*Uchoa et al., 2017*).

The implementation of RSSD used for the experiments in this article are set on a Euclidean Plane; therefore, distance and time are considered equivalent and will be used interchangeably. Below, the term *instance* is used to describe a distinct set of vertices and their locations on different graphs. These vertices are interchangeably referred to as nodes; commonly, in VRP literature, nodes are written as customers. For reference, Table 2 contains symbols and definitions that are used throughout the following subsections.

## Algorithm overview

The RSSD algorithm utilizes a greedy approach to identify *time* and *capacity* compliant delivery routes for the distribution of MCMs to PODs. The RSSD algorithm attempts to solve a version of the vehicle routing problem (VRP) that can be described as an open, capacitated, homogeneous multi-vehicular, MCM distribution, and multi-depot delivery

VRP. The RSSD algorithm guarantees that the generated delivery routes are both time and capacity compliant without requiring the number of routes (or number of vehicles) as input. RSSD has been created to *satisfice* the constraints of MCM delivery during the planning and activation stages. Although, in this article, we focus on the use of RSSD after a deliberate or accidental release of biochemicals, the routing algorithm can be used in situations where: the number of vehicles required is unknown, a time and capacity constraint must be met, and a need exists for a quick convergence (*i.e.*, solutions in seconds). A version of this algorithm was previously reported in *Urbanovsky (2018)*. A flow chart of the algorithm is included below, Fig. 2. Appendix A includes further modes of explaining the algorithm: (1) RSSD in algorithm notation; and (2) text version of the flowchart, Fig. 2. The version of RSSD as described below and used for running the experiments has been implemented in a Python3 Jupyter Notebook and published at www.doi.org/10.17605/OSF.IO/5H3GS.

The RSSD algorithm performs under five assumptions: (1) all demand can be met; (2) capacity of the vehicles is homogeneous; (3) vehicles leave the depot at the same time; (4) routes have no minimum time; and (5) vehicles do not return to the depot. Due to these assumptions, RSSD can rely on creating solutions that will serve all the demand of the PODs, use the same capacity and time constraint for all routes, and create open routes.

Consider a graph $G$ defined on a Euclidean plane $G = (V, E)$, composed of a vertex set of PODs $V = \{v_1, v_2, \ldots v_n\}$ which includes a depot set, $p \subset V$, and a directed, symmetrical edge set from which a distance matrix, $D$, is made. The distance matrix, $D$ contains all distances between each pair of nodes, $D_{ij} = t(v_i, v_j)$. In a non-Euclidean graph implementation, the edges may not be symmetrical because of one-way roads and/or inclusion of traffic conditions. For every $v_i$, there is a demand $c(v_i)$, except for the depot/s $p$, where the demand is 0, $c(p) = 0$. When implemented within a biomedical emergency context, each node's demand can be equal to the population of the catchment area that is being served by the POD. The time constraint for the length of the route, $\tau$, and the capacity constraint of the vehicles carrying the supplies, $\psi$, are also given. In the RE-PLAN implementation, the time constraint is chosen by the public health preparedness planners, and can be a range of time, where for every value within the range, solutions will be produced. The capacity constraint in the RE-PLAN implementation is the amount of individual MCMs an eighteen-wheeler can carry, which for an anthrax outbreak is around 211,200 doses. The output of the algorithm is a set of ordered lists made up of a number of $v \in V$, called routes $R = \{r_1, r_2, \ldots r_m\}$ where $r_1 = v_1, v_2, \ldots v_k$. For each $r_m$, there is the distance that the route covers $t(r_m)$, and the total demand being served on the route is $c(r_m)$. This graph and its variables are input of the RSSD algorithm, which is depicted in Fig. 2.

The first step of the algorithm is to check if routes can be created with the given time constraint, $\forall v_i \in \{V \backslash p\}, max(t(v_i, p)) \leq \tau$. This checks to see if the maximum distance from the vertices to their closest depot is less than the time constraint. If one of the distances from a vertex to the closest depot is greater than the time constraint, routes cannot be created and RSSD will return an empty set. After this initial check, another check on the capacity constraint is executed for every node. For every node that has a greater

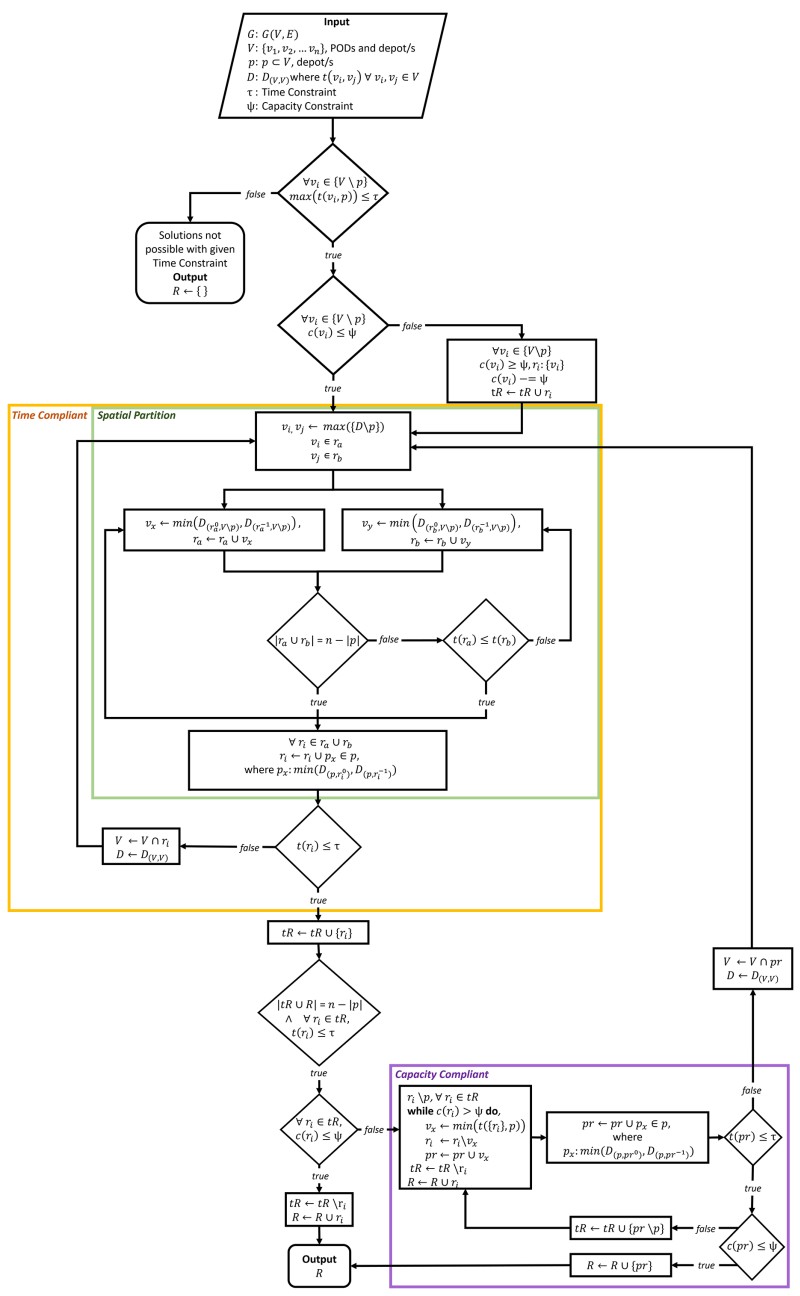

**Figure 2 Flowchart of RSSD algorithm.**

demand than the capacity constraint, $\forall v_i \in \{V \backslash p\}, c(v_i) \geq \psi$, a route will be created from the depot to that node $R \cup r_i : v_i$. The demand needed to be met at that node is then updated $c(v_i) - = \psi$.

The *Time Compliant* (yellow) section ensures created routes are within the given time constraint $\tau$ by surrounding a function, *Spatial Partition* (green), with a while loop that runs until all routes are time compliant $\forall r_i \in R, t(r_i) \leq \tau$. The *Spatial Partition* function, first identifies the nodes with the maximum distance from one another, $v_i, v_j = max(D \backslash p)$, not including the depot/s. These two nodes $v_i, v_j$ are selected as the first nodes of two

routes, $v_i \in r_a$ and $v_j \in r_b$. This metaheuristic was implemented because it is unlikely that the two furthest apart nodes will be on the same route given a time constraint. Next, these two routes, each composed of one node, will add the closest node to itself. The route with the shortest distance $t(r_a) \leq t(r_b)$ will then add the next closest node to the either end of the route $min(D_{r_c^0, V}, D_{r_c^{-1}, V})$. The routes will iteratively add the closest node to either end of the route, with the route that is currently the shortest being the one to add a node next. This continues until all of the nodes have been added to one of the routes, $|r_a \cup r_b| = n - |p|$. When all nodes are added to either $r_a$ or $r_b$, the closest depot will be added to the closest end of the routes. Then $t(r_a)$ and $t(r_b)$ will be checked against the time constraint, $\tau$. If either route is over $\tau$ that route will be sent through the *Time Compliant* section once again, but only with the nodes within the route as part of the distance matrix $D$ and vertex set $V$ is updated to only contain nodes from the route $r_i$. This *Time Compliant* section recursively repeats until every route meets the given time constraint.

The *Capacity Compliant* (purple) portion of the algorithm takes in the time compliant routes and ensures they meet the capacity constraint, $\psi$. The set of routes $R$ that do not meet the capacity constraint, $c(r_i) \geq \psi$ have the closest nodes to the depot pruned $v_x = min(t(r_i, p))$ until $r_i$ meets the capacity constraint. The nodes that are pruned from the non-capacity compliant routes are added to a new route, $pr$. This metaheuristic is implemented because pruning nodes that are closest to the depot will more likely result in a route that is time compliant. If the new route of pruned nodes do not meet the time constraint, it is sent through the *Time Compliant* portion of the algorithm until every route made up of the pruned nodes meets the time constraint. Then the capacity constraint is again checked. If the routes composed of the pruned nodes do not meet the capacity constraint, these routes will go through the capacity compliant portion again. This process is repeated until all routes are time and capacity compliant. When all routes, $R$, are compliant the final solutions are returned.

The RSSD algorithm is composed of two main sections that separately ensure time and capacity compliance. The time compliance portion relies on the metaheuristic that nodes furthest apart in a graph, are less likely to meet a given time constraint when in the same route. This section of the algorithm can also be thought of as clustering nodes together to identify nodes for routes and could potentially be used in other domains when the number of routes is not provided. The capacity portion of the algorithm relies on the metaheuristic that the nodes closest to the depot when pruned from non-capacity compliant routes will more likely create a time compliant route. RSSD is a simple, quick, and greedy algorithm that was designed to fit the needs of public health preparedness planners by providing the number of routes and creating solutions that are time and capacity compliant.

### Benchmarking methods

To benchmark the quality of RSSD solutions, we utilize best known solutions from the CVRPLIB (*Uchoa et al., 2017*) as if they are the *ground truth*. We were unable to find any algorithms that solve the exact same problem as RSSD, therefore, benchmarking the RSSD algorithm to such algorithms would not yield a fair comparison. Instead of comparing the RSSD algorithm to another algorithm, we endeavor to benchmark the solutions from

RSSD to a best known set of solutions within the CVRP domain so to evaluate the overall performance of the RSSD's route solutions.

As discussed in *Related Works*, the CVRPLIB repository contains regularly used CVRP benchmarking datasets. Many of the solutions have a designated best known solution within the problem space of CVRP. The algorithms that produce the best known solutions create closed, primarily capacity-optimized, and secondly distance-optimized routes. Although RSSD produces open routes and does not optimize for capacity nor time, the VRP that RSSD is ultimately solving is a CVRP, and thus using route solutions within this problem space is relevant to compare the solution results.

We focus our analysis on one dataset from CVRPLIB, which is listed as 'Uchoa et al. (2014)' and also designated with an 'X' in the online repository. This dataset was published along with the CVRPLIB repository in *Uchoa et al. (2017)*. Every instance in dataset 'X' is on a [0, 1,000] × [0, 1,000] grid. The locations of the depot are varied from central (C), at the origin (E), or randomly placed in the grid (R). The positions of the nodes are either random (R), clustered (C), or partially random and clustered (RC). The demand distributions for the nodes are uniform (U); small values between 1–10 or 5–10; large values between 1–100 or 50–100; dependent on quadrant (Q); or many small values and a few large values (SL). The combinations of these attributes in the datasets are split close to uniformly across the 100 instances in the dataset. Each instance in the dataset includes a capacity constraint, the locations of all nodes (including the depot), and the demands at each node.

The first portion of the benchmarking section reports CPU time between algorithms that engage in the NP-Hard task of identifying an optimal solution and RSSD which instead *satisfices* for the above described instances. Although, we find that comparing algorithms not helpful in our benchmarking process, we felt that comparing the CPU times of the algorithms reported in *Uchoa et al. (2017)* and RSSD as evidence for choosing a *satisficing* approach over engaging with the NP-Hard nature of the of the problem. In *Uchoa et al. (2017)*, they report three algorithms' CPU time in minutes for each of the instances in dataset 'X.' These three algorithms are iterated local search-based matheuristic (ILS-SP), the unified hybrid genetic search (UHGS), and branch-cut-and-price (BRS). We will combine their table with the RSSD process run-time in the 'Results' section. The ILS-SP and UHGS tests were conducted on Xeon CPU with 3.07 gigahertz and 16 gigabytes of RAM on an Oracle Linux Server 6.4 as reported in *Uchoa et al. (2017)*. UHGS was given a high termination criterion of up to 50,000 consecutive iterations without improvement (*Uchoa et al., 2017*). The BCP tests were run on a single core of an Intel i7-3960X 3.30 gigahertz processor with 64 gigabytes RAM as reported in *Uchoa et al. (2017)*. The RSSD tests were conducted on an Intel i5-10500 3.10 gigahertz processor, with 32 gigabytes of RAM. The implementation of RSSD for this article was not parallelized, therefore, all processes were conducted on one core, see notebook published at at www.doi.org/10.17605/OSF.IO/5H3GS.

The goal of our research is to benchmark the solutions of RSSD to the published 'X' best known solutions from CVRPLIB (*Uchoa et al., 2017*). Two aspects of the best known

solutions need to be mitigated: the closed routes in the 'X' solution set and the lack of a given time constraint. To fix the closed routes attribute of the solutions, we calculated the final metrics by removing the longest edge that joins the route to the depot, thus making the routes open. To resolve the absence of a time constraint for the instances, we use the best known solution's longest route distance (again excluding the longest segment to the depot) as the time constraint, $\tau$.

To measure the quality of the best known and RSSD solutions, we utilize three metrics that quantify consequential components in the domain of MCM dispersal at the time of plan activation. When evaluating the RSSD solution set, it was particularly imperative that we analyzed whether the utilization of resources (*e.g.*, vehicles) was excessive as a result of RSSD's lack of optimization. We assumed that the utilization of resources would be higher using the non-optimized, satisficing RSSD algorithm to produce the routes but we endeavored to find out how much. Using our three metrics, we quantify this trade-off for not engaging with and/or sidestepping this NP-Hard problem.

The *UnCap* metric is a measure of the percentage of unused capacity in relation to the sum of the demand of all nodes in the instance. It is an indication of potentially excessive routes/vehicles in the solution than necessary.

$$UnCap = \frac{(|R| * \psi) - c(V)}{c(V)} \tag{1}$$

where $R$ is the set of routes in a solution, $\psi$ is the capacity constraint, $V$ is the set of nodes, and $c(V)$ is the demand of all nodes. RSSD does not optimize for capacity, but it does perform certain metaheuristics in order to make better-informed greedy decisions. The closer the metric is to 0 indicates the less unused capacity in relation to the sum of all demands. Even with these meta-heuristics, we expected to see a significant difference between the best known and RSSD solutions in terms of unused capacity.

The *RngMax* metric measures the range between the solution's route distances and evaluates the variance in route sizes. All PODs in a region must open at the same time in order to reduce the possibility of over-crowding at the PODs that open first. This amount measures potential excess resources of guards at POD locations. RngMax is a ratio between the maximum route distance and minimum route distance over the maximum route distance.

$$RngMax = \frac{max(t(R)) - min(t(R))}{max(t(R))} \tag{2}$$

where $R$ is the set of routes in a solution and $t(R)$ is route distance. A high value of the RngMax metric indicates a large variance in route lengths. If some routes are significantly longer than others, guards may be required for longer for the PODs that have already received deliveries on shorter routes. When routes have similar distances, PODs along each route will be served around the same time. The closer the RngMax score is to 0, the less distance in between the route ranges.

The *NoR* metric is a ratio of the solution's number of routes over the total number of nodes. This is similar to unused capacity in that it measures excessive use of resources, but instead with focusing on the number of nodes in relation to number of routes.

$$NoR = \frac{|R|}{n} \tag{3}$$

where $R$ is the set of routes and $n$ is the total number of nodes. This ratio is a measure of how much of a route per node is in the instance, and also allows for uniform comparison across all instances in a dataset. Dividing by the total number of nodes in an instance normalizes the metric and allows for comparison across all instances. The lower the number of routes the fewer resources (vehicles, *etc.*) that need to be reserved in the case of plan activation. A smaller value of this metric for an instance indicates that fewer routes are needed to serve the number of nodes.

For further evaluation, we will compare metrics across a set of characteristic attributes of the instance. These attributes of an instance include: depot location, node positioning, demand distribution, number of nodes in an instance, time constraint length, capacity constraint size, and the sum of all node demands. This analysis is in order to quantify whether there are significant differences of quality of the solutions produced when the graph has a specific attribute. These attributes are features of each dataset (location of nodes/depot, demand distribution, *etc.*) that can be binned across a continuum and compared across all instances.

The difference between the mean metric (UnCap, RngMax, NoR) for the best known and RSSD solutions is taken to indicate the compared quality of the solutions for that metric. When the difference is negative, the best known solutions have a lower value for that metric, and when the difference is positive the RSSD metric is lower. A difference of 0 indicates the RSSD and best known solutions perform the same.

For the mean difference values, a trendline is computed to allow for quantitative analysis of potential patterns across the attributes continuum (increase in the attribute quantity or randomness). This trendline is a simple least squares fitting line. If the coefficient of determination value or $R^2$ is high, this indicates that the data fits the linear regression model. If $R^2$ value is high *and* the slope of the trendline is 0, this indicates that the RSSD and the best known set of solutions do not behave differently based on the continuum of the attribute.

To further compare the means, we perform statistical tests that can indicate if the means of the dependent variables (the attribute categories) and the independent variables (the mean difference for the three metrics) perform significantly differently. The One-Way ANOVA $p$-values were used when Levene's test results were not significant; the Welch ANOVA is used when the result of Levene's test were significant. If there is significance between the variables based on the $p$-values from the ANOVA tests, the Games-Howell posthoc test was performed. The Welch ANOVA and the Games-Howell posthoc test were chosen for their robustness against datasets that violate the homogeneity of varianc and equal sample sizes. With only 100 instances in the 'X' dataset and fewer than 20 instances in each attribute category, these results may be affected by the law of small numbers. These

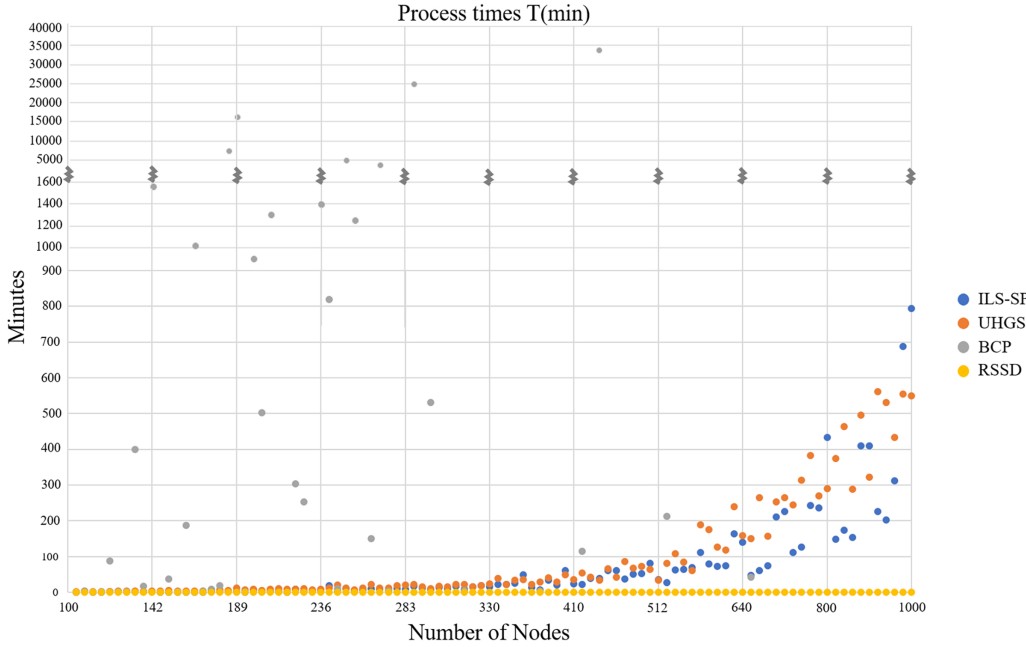

**Figure 3 Four algorithms' CPU times in minutes (y-axis) over all instances by number of nodes (x-axis).**

statistical tests were performed on SPSS Version 28 and the queries utilized and every instance's results are published at www.doi.org/10.17605/OSF.IO/5H3GS.

## RESULTS

First, we compare RSSD CPU processing times to those reported in *Uchoa et al. (2017)*, for ILS-SP, UHGS, and BRS, see Fig. 3. These three algorithms engage in the NP-Hard nature of the problem of attempting to generate optimal solutions and take longer to produce accepted solutions compared to the RSSD algorithm. The stopping criteria that was chosen for UHGS (*Uchoa et al., 2017*) appears excessively high (up to 50,000 consecutive iterations without improvement) and therefore the CPU time for this algorithm could be considered inflated. For all instances, RSSD produces a solution (ranging from 101 to 1,001 nodes) in less than a minute. See Appendix B for two more charts on the processing times (Figs. B1 and B2).

Across all instances, the distribution of our devised three metrics, UnCap, RngMax, and NoR, are reported in Fig. 4. See Appendix C for Table C1 of the mean, min, max, and standard deviations. The best known solution's mean *UnCap* metric is ≈0.03% capacity unused of the instance's total demand; comparatively to the RSSD solutions, which have an average unused capacity ≈25%. This result is not surprising in that the best known solutions are created by algorithms for which optimizing for capacity usage is priority. The results of the mean *RngMax* metric indicate that the RSSD solutions have a smaller variance among the route lengths in a solution than that of the best known solution set. The mean *NoR* metric's results are that the best known solutions have ≈0.127 routes per node on average, and the RSSD have ≈0.152 routes per node. The NoR difference of

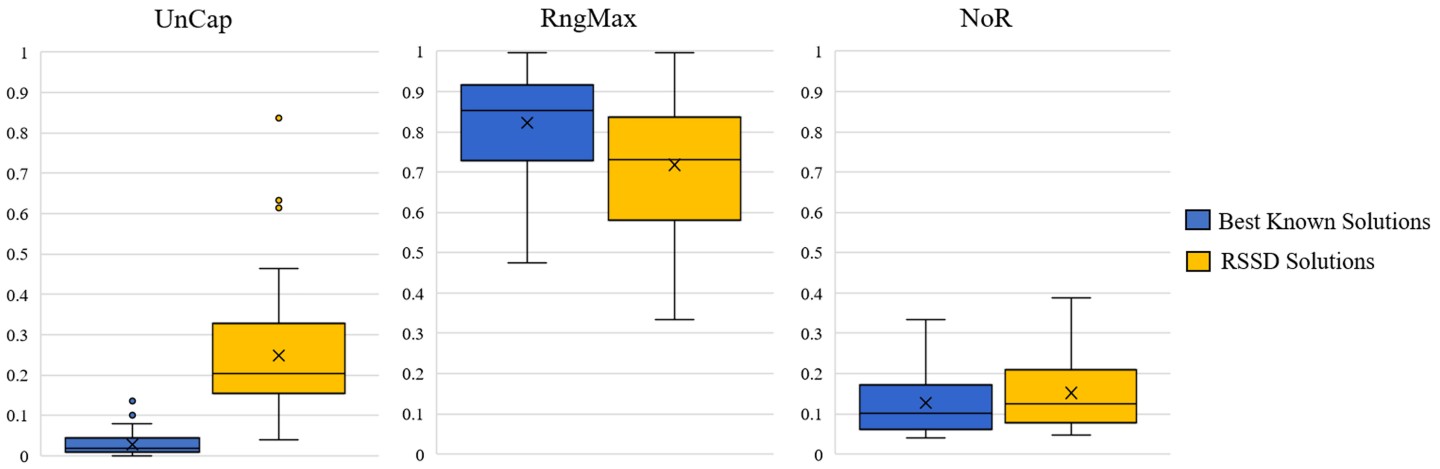

**Figure 4 Distribution across the three metrics for all instances.**

≈0.026 overall is not excessive and indicates that although the RSSD algorithm does not engage in optimization and has a higher UnCap score, there is not a large difference in the number of routes per solution over all the instances. For each of the metrics, the RSSD solutions have a much wider variance in scores in comparison to that of the best known solutions'. This is indicative of the greediness of RSSD and was expected. For the results of each variable and metric for each instance, see the document published at www.doi.org/10.17605/OSF.IO/5H3GS.

The next step is to evaluate whether or not there is a trend over the course of an increase of attribute (number of nodes, time constraint, capacity constraint, randomness *etc.*) and the quality of the solution. Figures 5 and 6 contain the distributions for the best known and RSSD solutions for each attribute for each metric as well as the trendline of the plotted differences (best known minus RSSD) of the solutions means. The columns in the graphs represent categorical groupings of nodes based on the attributes of the instances. Table 3 contains the grouping label (*e.g.*, 1–9), the criteria for grouping (*e.g.*, *range*: 101–190 or *label*: center) and the number (*n*) per grouping. The quality of the solution is defined as the similarity of trends between the best known and the RSSD solutions.

The trendline's slopes across all of the attributes and metrics stayed close to 0 in Figs. 5 and 6. This indicates that as the increase in the attributes' value (*e.g.*, the more nodes in an instance) or the increase in randomness (*e.g.*, depot position), there were not significant trends in the RSSD solutions quality in comparison to that of the best known solutions. The highest $R^2$ values for the trendlines were: the attributes depot positioning (0.98), and node positioning (0.96) by the *UnCap* metric; capacity constraint (0.6) by RngMax and; time constraint (0.9) and node positioning (0.89) by *NoR*. This indicates that the slope of the trendline is more accurate in representing the trend of these attributes. The depot positioning (0.037) and node positioning (0.056) by the *UnCap* metric had the largest slope values. For depot positioning, as the location became more random, the RSSD and best known solutions converged closer in *UnCap*. As the node positioning became more

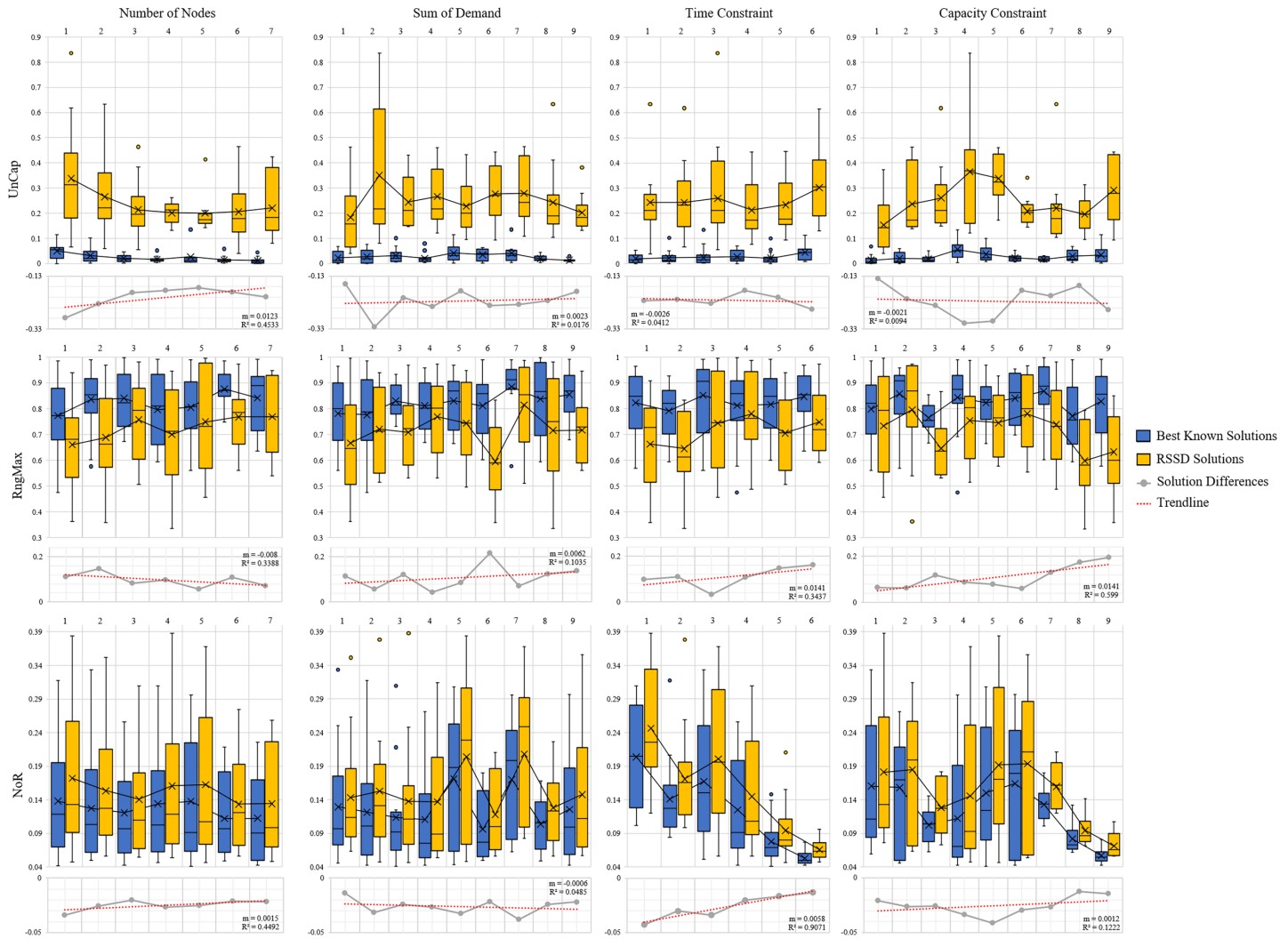

**Figure 5 Compare means graphs for *number of nodes, time constraint, capacity constraint*, and *sum of demands*.**

random, the RSSD had a larger mean value in *UnCap* and the difference trended downward, the quality of the RSSD solutions diminished in relation to the randomness of node positions in instances.

The last step in benchmarking is to evaluate if there are any significant differences in quality between groupings among attributes for each of the difference metrics. The null hypothesis is that there are no differences among the attribute groupings therefore there should not be any significant difference among the groups. The three difference metrics underwent multiple statistical tests as explained in *Benchmarking Methods*. The results when there are significant differences between groups are shown in Table 4.

There were 15 pairs of groupings that were significantly different out of 405 total paired groupings. The *UnCap* metric had significant differences between some groups by capacity constraint, node positioning, and demand distribution. This indicates that for these

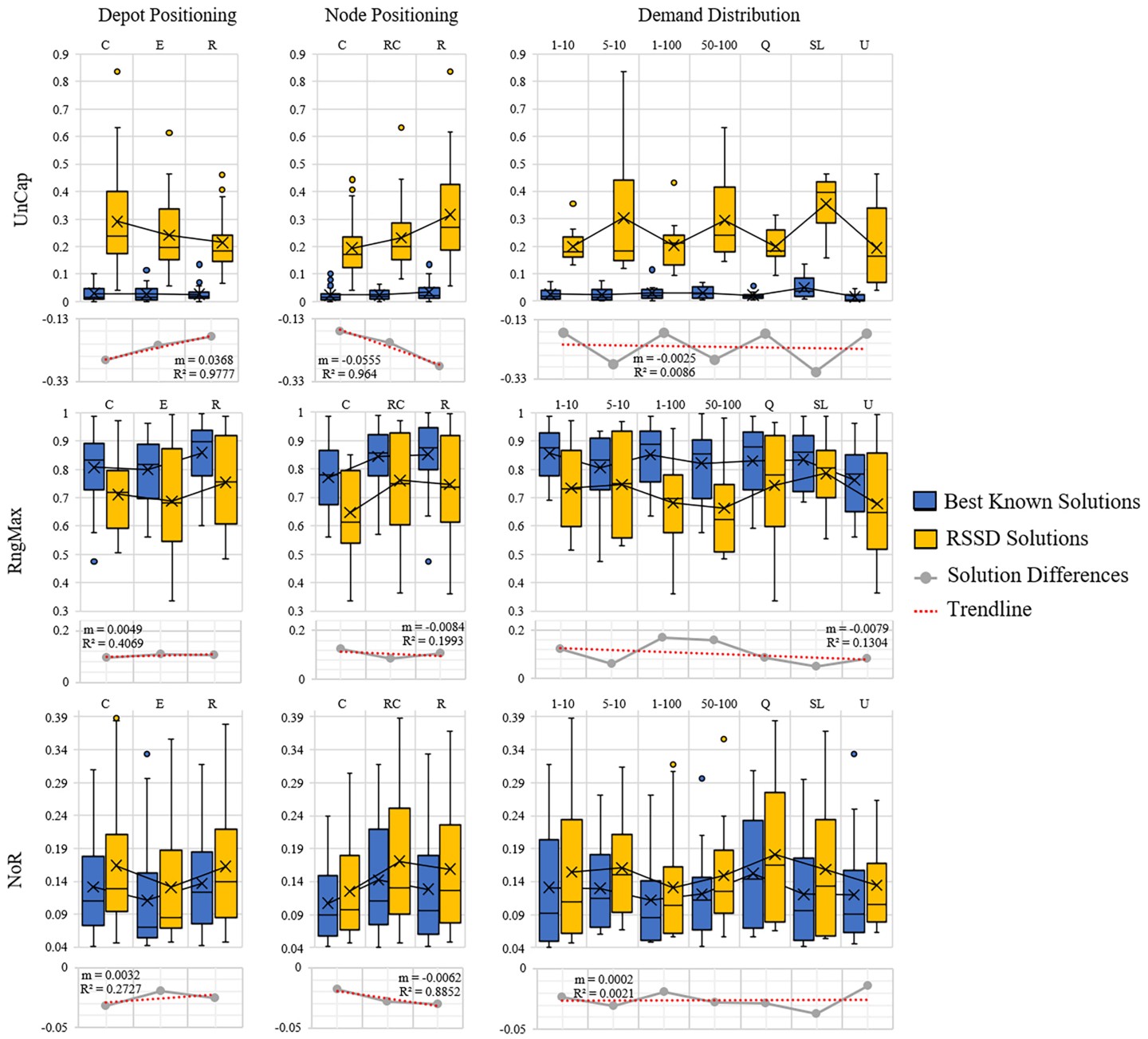

**Figure 6  Compare means graphs for *depot positioning, node positioning,* and *demand distribution*.**

attributes, the optimal and RSSD behaved significantly different within these groupings. The *RngMax* had significant difference between one pair within the attribute Time Constraint. The *NoR* metric had the most pairs of groupings that behaved differently among the three metrics, with difference among groupings in Time Constraint, Capacity Constraint, Depot Positioning, and Node Positioning. From this, we can expect RSSD results to behave differently than a given *ground truth* if the instances meet these criteria.

**Table 3 Attribute labels and ranges.**

| Attribute | Label | 1 | 2 | 3 | 4 | 5 | 6 | 7 | 8 | 9 |
|---|---|---|---|---|---|---|---|---|---|---|
| Number of Nodes | range | 101–199 | 200–299 | 300–399 | 400–499 | 500–599 | 600–799 | 800–1,001 | | |
| | n | 21 | 22 | 15 | 10 | 9 | 12 | 11 | | |
| Sum of Demands | range | 119–799 | 800–1,499 | 1,500–2,999 | 3,000–4,999 | 5,000–9,999 | 10,000–14,999 | 15,000–19,999 | 20,000–34,999 | 35,000–67,057 |
| | n | 14 | 10 | 12 | 13 | 12 | 8 | 9 | 12 | 10 |
| Time Constraint | range | 780–999 | 1,000–1,199 | 1,200–1,499 | 1,500–1,699 | 1,700–1,999 | 2,000–2,850 | | | |
| | n | 15 | 16 | 19 | 23 | 17 | 10 | | | |
| Capacity Constraint | range | 19–March | 20–49 | 50–99 | 100–139 | 140–199 | 200–399 | 400–599 | 600–899 | 900–1,816 |
| | n | 15 | 11 | 9 | 12 | 10 | 13 | 9 | 10 | 11 |
| **Depot Positioning** | label | **Center** | **Origin (E)** | **Random** | | | | | | |
| | n | 32 | 34 | 34 | | | | | | |
| **Node Positioning** | label | **Clustered** | **Clustered/ Random** | **Random** | | | | | | |
| | n | 32 | 34 | 34 | | | | | | |
| **Demand Distribution** | label | 1–10 | 5–10 | 1–100 | 50–100 | **Quadrant** | **Many Small, Few Large** | **Uniform** | | |
| | n | 14 | 14 | 14 | 14 | 14 | 14 | 16 | | |

**Table 4 Statistical significance of difference of means between groupings.**

| Metric | Attribute | Group I | Group J | Mean difference | Sig. | 95% Confidence interval | |
|---|---|---|---|---|---|---|---|
| | | | | | | Lower bound | Upper bound |
| UnCap | Capacity Constraint | 1 | 5 | 0.161 | 0.009 | 0.031 | 0.292 |
| | | 5 | 8 | −0.135 | 0.036 | −0.264 | −0.006 |
| | Node Positioning | C | R | 0.111 | 0.003 | 0.034 | 0.188 |
| | Demand Distribution | 1–10 | SL | 0.133 | 0.003 | 0.039 | 0.227 |
| | | 1–100 | SL | 0.132 | 0.004 | 0.034 | 0.230 |
| | | Q | SL | 0.128 | 0.004 | 0.033 | 0.223 |
| RngMax | Time Constraint | 3 | 5 | −0.114 | 0.010 | −0.208 | −0.020 |
| NoR | Time Constraint | 1 | 6 | −0.030 | 0.047 | −0.059 | 0.000 |
| | | 2 | 6 | −0.017 | 0.027 | −0.033 | −0.002 |
| | | 3 | 5 | −0.017 | 0.046 | −0.034 | 0.000 |
| | | 3 | 6 | −0.021 | 0.002 | −0.035 | −0.006 |
| | Capacity Constraint | 2 | 8 | −0.014 | 0.010 | −0.025 | −0.003 |
| | | 5 | 8 | −0.029 | 0.036 | −0.057 | −0.002 |
| | Depot Positioning | C | E | −0.012 | 0.030 | −0.023 | −0.001 |
| | Node Positioning | C | R | 0.012 | 0.016 | 0.002 | 0.023 |

# CONCLUSION

The RSSD algorithm employs a *satisficing* strategy to identify solutions that meet the given time and capacity constraints. The algorithm is quick and does not require the number of vehicles/routes as input. With its speed and low algorithmic complexity, it is well suited for

the problem for which it was designed in the decision support system, RE-PLAN. To quantify the lack of engagement in the NP-Hard problem of finding an optimal solution, we devise three metrics that are relevant within domain. These three metrics take into account components of the solutions that could result in higher resource utilization. The UnCAP and NoR metrics quantify a potential overuse of vehicles for routes. The RngMax metric quantifies a potential excess of guards at POD locations that have already received resources. These three metrics are appropriate for quantifying the quality of RSSD in respect to the ground truth as they assess excess use of resources in appropriate ways.

Using these metrics allowed us to benchmarking the RSSD solutions to best known solutions in a context specific way. This benchmarking is essential in our ability to understand if the RSSD algorithm produces consistent and acceptable solutions for use. The methods of benchmarking created and described above are able to address whether or not this is the case. The RSSD solutions on average have less variance in route length (lower RngMax scores) than that of the best known solutions, which is beneficial for PHPPs in timing the beginning of dispensing of MCMs across a region. The means for *UnCap* and *NoR* for the RSSD solutions were higher than that of the optimal, however, the quality of the solutions were consistent across the attributes of those instances. There were few grouping pairs that had significant differences, meaning that in general the RSSD and best known solutions performed similarly. The RSSD algorithm produces competitive solutions compared with the best known solutions, using the three devised domain-specific metrics.

A major limitation of this benchmarking method is that dataset 'X' contained only 100 curated instances (*Uchoa et al., 2017*). If more instances had the same curated qualities, we would be able to benchmark the RSSD solutions with more confidence. Another limitation resulting from the small dataset size, is the groupings for the instance attributes (*e.g.*, Time Constraint, Capacity Constraint, *etc.*) were at times small with 7–14 instances per grouping. These smaller groups could contain disproportionately certain other attributes, which ultimately could impact the scores of the solutions more so then that of the attribute grouping being compared.

Future work involves expanding the scope of the RSSD algorithm's applications, including its utilization in managing emergency deliveries for drones, planning routes for public transportation, and incorporating it into algorithms for locating depots. Another research direction involves investigating partitioning strategies for the initial spatial partitioning phase of the RSSD algorithm. Additionally, there is potential to devise additional metrics for assessing generated routes based on energy consumption or closure risk due to hazards.

### Funding

The authors received no funding for this work.

## Competing Interests

Armin R. Mikler is an Academic Editor for PeerJ. The authors declare that they have no competing interests.

## Author Contributions

- Emma L. McDaniel conceived and designed the experiments, performed the experiments, analyzed the data, performed the computation work, prepared figures and/or tables, authored or reviewed drafts of the article, and approved the final draft.
- Sampson Akwafuo analyzed the data, prepared figures and/or tables, and approved the final draft.
- Joshua Urbanovsky analyzed the data, authored or reviewed drafts of the article, proposed the RSSD algorithm in their dissertation, and approved the final draft.
- Armin R. Mikler conceived and designed the experiments, analyzed the data, authored or reviewed drafts of the article, and approved the final draft.

## Data Availability

The data and code is available at OSF: McDaniel, Emma L. 2023. "Benchmarking a Fast, Satisficing Vehicle Routing Algorithm for Public Health Emergency Planning and Response." OSF. June 10. DOI 10.17605/OSF.IO/5H3GS.

## Supplemental Information

Supplemental information for this article can be found online at http://dx.doi.org/10.7717/peerj-cs.1541#supplemental-information.

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
