# Peer review of "Benchmarking a fast, satisficing vehicle routing algorithm for public health emergency planning and response: “Good Enough for Jazz”"

_PeerJ Computer Science, doi:10.7717/peerj-cs.1541_

## Round 0.1 · original submission · Major Revisions

Please fully address all the concerns raised by both reviewers in the revised manuscript or justify if the corresponding comments are not accurate thus not addressed.

Reviewer 1 ·

Basic reporting

Abstract:
The authors give details of the proposed methods and experimental results in the abstract section without highlighting the contributions of this paper.

Background section:
The quality of Figure 1 should be improved.
It is unclear what the motivation is for using RSSD algorithms to solve constrained vehicle routing problems. How the RSSD algorithm is used in different stages, i.e. planning stage and plan activation, is unclear.

Related work section:
Avoid using long sentences, such as “Their assumptions…from the SNS” in lines 123-126, to improve the readability of the paper.
The format of citing references needs to be improved, e.g “Jaillet et al. (2016) Jaillet et al. (2016)” in line 148 and “Aldy Gunawan and Graham Kendall and Barry McCollum and Hsin-Vonn Seow and Lai Soon Lee (2021)” in line 160.
The title of Table 1 should be placed at the top.

Are all the algorithms listed in Table 1 heuristic algorithms? What is the difference between “optimizing” and “satisficing”? The motivation for selecting them as the comparison algorithms to RSSD is not given. It seems that the authors randomly picked up some of the related literature and listed them in this section without justification and summary.

Two datasets repositories, i.e. VRP-REP and CVRPLIB, are mentioned in this section. However, in the following experimental section, only CVRPLIB datasets are used.

Experimental design

Problem Description section:
In line 186, the authors claim that “The RSSD algorithm is a member of VRP variants”, which is strange. Actually, RSSD is the algorithm/method to solve VRP. It is not a type of VRP.

Figure 2 shows the flow chart of the RSSD algorithm; however, it is hard to follow since there are quite a lot of mathematical symbols and formulas. The authors are suggested to explain the whole flow chart in the text rather than just giving the flow chart figure.

The equations listed in this section should be numbered.

Validity of the findings

Result section:

The authors claim that three comparison algorithms (i.e. ILS-SP, UJGS, BRS) are optimal, which is inaccurate. It seems that the authors misunderstand the concept of optimal algorithms. ILS and GS are heuristic algorithms that cannot guarantee the optimality of the solutions.

Is the time comparison in Figure 3 fair? What are the computing platform, and stopping criteria of these four algorithms?

Conclusion section:
The conclusion section is most like a summary of the experimental results instead of a summary of the whole paper. The authors are suggested to rewrite the conclusion section to highlight the contributions of this paper instead of giving many details of the results.

Reviewer 2 ·

Basic reporting

This paper aims to evaluate the Receiving-Staging-Storing-Distributing (RSSD) algorithm for solving a specific instance of a multi-vehicular routing problem proposed by (Urbanovsky, 2018). The RSSD aims to produce satisfying solutions not optimal solutions for the targeting routing problem. It has been investigated in many works, such as (Urbanovsky, 2018) and (O’Neill et al., 2014) as reviewed in this paper. To benchmark the RSSD algorithm, the CVRP problem is taken as a case study to compare the RSSD solutions against the optimal/near-optimal solutions based on three metrics.

The literature has been reviewed in a relatively acceptable manner. The experimental results are almost structured and their evaluations and discussions are clear.

However, the contribution has not been well-explained and thus should be improved. It seems the RSSD algorithm and the problem model have not been developed any further in this article as far as I can see.

In addition, the article is not free of typo/grammar mistakes, e.g. Page 2, line 53, “algorith”, Page 5, line 213, “... 2014”. The legend of Figure 5 and Figure 6 is missing.

Experimental design

The article claims the importance of benchmarking the RSSD performance in understanding if the algorithm produces consistent and acceptable solutions to be used within the public health emergency context, which is new to the literature. However, it seems the results are not adequate to demonstrate the benchmarking in the following aspects.

1) the experimental study is conducted in the CVRP domain not in the public health emergency problem, while several widely-studied meta-heuristics (as reviewed in the article) have been investigated in more relevant scenarios. What are the additional constraints in the public health emergency context in addition to the standard CVRP?

2) Compared with only the optimal results is difficult to assess how well the RSSD algorithm performs. For example, the difference in terms of NoR between RSSD and the optimal results is 0.026 - how good is it? What is the difference between the optimal results against the results of other well-known meta-heuristics?

Validity of the findings

No comment.

---

## Round 0.2 · Minor Revisions

Please address the final minor comments from the review.

Reviewer 1 ·

Basic reporting

Thank you for taking the time to address my comments and suggestions. The manuscript looks much better now. I only have three more minor comments:

Abstract section:
Avoid using acronyms in abstracts.

Experimental design

I understand why only VRP-REP is not included in the experiment section now. The authors are suggested to explain this point somewhere in the paper, instead of just explaining it in the Rebuttal Letter.

Validity of the findings

Conclusion section:
The future work section appears to be underdeveloped and lacks sufficient thought. It relies on general descriptions, using phrases such as "similar method" and "similar manner." For instance, what does the statement "more algorithms that are within emergency response" mean? Providing specific examples would be helpful in clarifying the point.

Reviewer 2 ·

Basic reporting

No comment

Experimental design

No comment

Validity of the findings

No comment

---

## Round 0.3 · accepted · Accept

I'm happy to recommend accepting the revised paper.